# GLOCAL HYPERGRADIENT ESTIMATION WITH KOOPMAN OPERATOR

## ABSTRACT

Gradient-based hyperparameter optimization methods update hyperparameters using hypergradients, gradients of a meta criterion with respect to hyperparameters. Previous research used two distinct update strategies: optimizing hyperparameters using global hypergradients obtained after completing model training or local hypergradients derived after every few model updates. While global hypergradients offer reliability, their computational cost is significant; conversely, local hypergradients provide speed but are often suboptimal. In this paper, we propose *glocal* hypergradient estimation, blending "global" quality with "local" efficiency. To this end, we use the Koopman operator theory to linearize the dynamics of hypergradients so that the global hypergradients can be efficiently approximated only by using a trajectory of local hypergradients. Consequently, we can optimize hyperparameters greedily using estimated global hypergradients, achieving both reliability and efficiency simultaneously. Through numerical experiments of hyperparameter optimization, including optimization of optimizers, we demonstrate the effectiveness of the glocal hypergradient estimation.

## 1 INTRODUCTION

A bi-level optimization problem is a nested problem consisting of two problems for the model parameters $\boldsymbol{\theta} \in \mathbb{R}^p$ and the meta-level parameters called hyperparameters $\boldsymbol{\phi} \in \mathbb{R}^q$ as

$$\boldsymbol{\phi}^* \in \underset{\boldsymbol{\phi}}{\operatorname{argmin}} \, \tilde{\ell}(\boldsymbol{\theta}^*(\boldsymbol{\phi}); \tilde{\mathcal{D}}) \tag{1}$$

$$\text{such that} \quad \boldsymbol{\theta}^*(\boldsymbol{\phi}) \in \underset{\boldsymbol{\theta}}{\operatorname{argmin}} \, \ell(\boldsymbol{\theta}, \boldsymbol{\phi}; \mathcal{D}). \tag{2}$$

Its inner-level problem (Equation (2)) is to minimize an inner objective $\ell(\boldsymbol{\theta}, \boldsymbol{\phi}; \mathcal{D})$ with respect to $\boldsymbol{\theta}$ on data $\mathcal{D}$. The outer-level or meta-level problem (Equation (1)) aims to minimize a meta objective $\tilde{\ell}(\boldsymbol{\theta}, \boldsymbol{\phi}; \tilde{\mathcal{D}})$ with respect to $\boldsymbol{\phi}$ on data $\tilde{\mathcal{D}}$, and usually $\mathcal{D} \cap \tilde{\mathcal{D}} = \emptyset$. A typical problem is hyperparameter optimization (Hutter et al., 2019), where $\ell$ and $\tilde{\ell}$ are training and validation objectives, and $\mathcal{D}$ and $\tilde{\mathcal{D}}$ are training and validation datasets. Another example is meta learning (Hospedales et al., 2021), where $\ell$ and $\tilde{\ell}$ correspond to meta-training and meta-testing objectives. In the deep learning context, which is the main focus of this paper, $\boldsymbol{\theta}$ corresponds to neural network parameters, which are optimized by gradient-based optimizers, such as SGD or Adam (Kingma & Ba, 2015), using gradient $\nabla_{\boldsymbol{\theta}} \ell(\boldsymbol{\theta}, \boldsymbol{\phi})$.

Similarly, gradient-based bi-level optimization aims to optimize the hyperparameters with gradient-based optimizers by using hypergradient $\nabla_{\boldsymbol{\phi}} \tilde{\ell}(\boldsymbol{\theta}(\boldsymbol{\phi}))$ (Bengio, 2000; Larsen et al., 1996). Although this hypergradient is not always available, when it can be obtained or estimated, gradient-based optimization is efficient and scalable to even millions of hyperparameters (Lorraine et al., 2020), surpassing the black-box counterparts scaling up to few hundreds (Bergstra et al., 2011; 2013). As a result, gradient-based approaches are applied in practical problems that demand efficiency (Choe et al., 2023), such as neural architecture search (Liu et al., 2019; Zhang et al., 2021; Sakamoto et al., 2023), optimization of data augmentation (Hataya et al., 2022), and balancing several loss terms (Shu et al., 2019; Li et al., 2021).

Hypergradients in previous works can be grouped into two categories: global hypergradients and local hypergradients. Gradient-based bi-level optimization with global hypergradients uses the gradient

Figure 1: The schematic view of hypergradients. We want to use global hypergradient $\boldsymbol{h}_T$ to update hyperparameters, but it needs to wait for the completion of the entire training process. Updating hyperparameters $\boldsymbol{\phi}_s$ with a local hypergradient $\boldsymbol{h}_{s\tau}$ is efficient but may lead to suboptimal solutions. To leverage both advantages, we propose *glocal* **hypergradient estimation** that approximates global hypergradient $\boldsymbol{h}_T$ using a local hypergradient trajectory $\boldsymbol{h}_t$ for $t \in [(s-1)\tau + 1, \ldots, s\tau]$, enabling to update $\boldsymbol{\phi}_s$ using $\boldsymbol{h}_T$ efficiently.

of the meta criterion value after the entire model training completes with respect to hyperparameters (Maclaurin et al., 2015; Micaelli & Storkey, 2021; Domke, 2012). This optimization can find hyperparameters to minimize the final meta criterion, but it is computationally expensive because the entire training loop needs to be repeatedly executed. Contrarily, gradient-based hyperparameter optimization with local hypergradient leverages hypergradients of the loss values after every few iterations of training and updates hyperparameters on-the-fly (Franceschi et al., 2017; Luketina et al., 2016). This strategy can optimize model parameters and hyperparameters alternately and achieve efficient hyperparameter optimization, although the obtained hypergradients are often degenerated because of "short-horizon bias" (Wu et al., 2018; Micaelli & Storkey, 2021).

In this paper, we propose *glocal hypergradient* that leverages the advantages and avoids shortcomings of global and local hypergradients. This method estimates global hypergradients using a trajectory of local hypergradients, which are used to update hyperparameters greedily (see Figure 1). This leap can be achieved by the Koopman operator theory (Koopman, 1931; Mezić, 2005; Brunton et al., 2022), which linearizes a nonlinear dynamical system, to compute desired global hypergradients as the steady state from local information. As a result, gradient-based bi-level optimization with glocal hypergradients can greedily optimize hyperparameters using approximated global hypergradients. We verify its effectiveness in numerical experiments, namely, hyperparameter optimization of optimizers and data reweighting.

Our specific contributions are as follows:

1. We propose the *glocal* hypergradient estimation that combines the advantages of global and local hypergradient techniques. Specifically, it leverages the Koopman operator theory to predict the global hypergradients from local information, thereby retaining the efficiency of local methods while capturing global information.

2. We provide a comprehensive theoretical analysis of the computational complexities involved in the glocal approach, revealing its computational efficiency compared to the global method. Additionally, we present an error bound quantifying the accuracy of the proposed glocal estimation relative to the actual global hypergradient.

3. We numerically demonstrate that the proposed glocal hypergradient estimation achieves performance comparable to global approach while maintaining the efficiency of the local method.

## 2 BACKGROUND

### 2.1 GRADIENT-BASED BI-LEVEL OPTIMIZATION

Gradient-based bi-level optimization solves the outer-level problem, Equation (1), using gradient-based optimization methods by obtaining hypergradient $\nabla_{\boldsymbol{\phi}} \tilde{\ell}(\boldsymbol{\theta}^*(\boldsymbol{\phi}))$.

When the inner-level problem (Equation (2)) involves the training of neural networks, which is our main focus in this paper, computing its minima is infeasible. Thus, we instead truncate the original

inner problem to the following $T$-step optimization process:

$$\phi^* \in \underset{\phi}{\arg\min} \; \tilde{\ell}(\boldsymbol{\theta}_T(\phi); \tilde{\mathcal{D}}) \tag{3}$$

$$\text{such that } \boldsymbol{\theta}_{t+1}(\phi) = \Theta(\boldsymbol{\theta}_t, \phi; \mathcal{D}), \tag{4}$$

for $t = 0, 1, \ldots, T-1$ and $\boldsymbol{\theta}_0$ is randomly initialized neural network parameters. $\Theta$ is a gradient-based optimization algorithm, such as $\Theta(\boldsymbol{\theta}_t, \phi; \mathcal{D}) = \boldsymbol{\theta}_t - \eta \nabla_{\boldsymbol{\theta}} \ell(\boldsymbol{\theta}_t, \phi; \mathcal{D})$ in the case of vanilla gradient descent, where $\eta$ is a learning rate and can be an element of $\phi$.

Similarly, we focus on the case that Equation (3) is also optimized by iterative gradient-based optimization

$$\phi_{s+1} = \Phi(\boldsymbol{\theta}_T, \phi_s; \tilde{\mathcal{D}}) \quad \text{such that } \boldsymbol{\theta}_{t+1}(\phi) = \Theta(\boldsymbol{\theta}_t, \phi_s; \mathcal{D}). \tag{5}$$

where $s = 0, 1, \ldots, S-1$. The optimization algorithm $\Phi$ in Equation (5) adopts hypergradient $\nabla_{\phi} \tilde{\ell}(\boldsymbol{\theta}_T(\phi))$, which is referred to as *global* hypergradient (Maclaurin et al., 2015; Micaelli & Storkey, 2021; Domke, 2012). This design requires $T$-iteration model training $S$ times, which is called non-greedy (Micaelli & Storkey, 2021). A greedy approach that alternately updates model parameters and hyperparameters is also possible: hyperparameters are updated every $\tau \; (< T)$ iteration by hypergradients obtained by a playout until the $T$-th model update $\nabla_{\phi_s} \tilde{\ell}(\boldsymbol{\theta}_T(\phi_s))$. In both cases, gradient-based bi-level optimization using global hypergradients requires a computational cost of $O(ST)$, which is computationally challenging for a large $T$.

Instead of waiting for the completion of model training to compute global hypergradient, *local* hypergradient $\nabla_{\phi_{s-1}} \tilde{\ell}(\boldsymbol{\theta}_{s\tau}(\phi_{s-1}))$ obtained every $\tau$ iteration can also be used to greedily update $\phi$ (Franceschi et al., 2017; Luketina et al., 2016; Wu et al., 2018). This relaxation replaces Equations (3) and (4) as

$$\phi_{s+1} = \Phi(\boldsymbol{\theta}_{(s+1)\tau}, \phi_s; \tilde{\mathcal{D}}) \quad \text{such that } \boldsymbol{\theta}_{t+1}(\phi) = \Theta(\boldsymbol{\theta}_t, \phi_s; \mathcal{D}), \tag{6}$$

for $t \in \mathcal{I}_s = [(s-1)\tau, (s-1)\tau + 1, \ldots, s\tau - 1]$. By setting $\tau \approx T/S$, that is, $S \approx T/\tau$, this approach approximately optimizes $\phi$ in an $O(T)$ computational cost. A downside of this approach is that the local hypergradients may be biased, especially when the inner optimization involves stochastic gradient descent (Wu et al., 2018; Micaelli & Storkey, 2021).

## 2.2 COMPUTATION OF HYPERGRADIENTS

To compute such global or local hypergradients, a straightforward approach is to differentiate through the $T$- or $\tau$- step optimization process (Finn et al., 2017; Grefenstette et al., 2019; Domke, 2012). This unrolling approach is applicable to any differentiable hyperparameters. Yet, it suffers from large memory requirements, $O(Tp)$ for the global one, when reverse-mode automatic differentiation is used. This challenge may be alleviated using forward-mode automatic differentiation (Micaelli & Storkey, 2021; Franceschi et al., 2017; Deleu et al., 2022) or truncated backpropagation (Shaban et al., 2019). Moreover, differentiating through long unrolled computational graphs suffers from gradient vanishing/explosion, limiting its applications.

Alternatively, implicit differentiation can also be used with an assumption that $\boldsymbol{\theta}_T$ reaches close enough to a local optimum. The main bottleneck of this approach is the computation of inverse Hessian with respect to model parameters (Bengio, 2000), which can be bypassed by iterative linear system solvers (Pedregosa, 2016; Rajeswaran et al., 2019; Blondel et al., 2021), the Neumann series approximation (Lorraine et al., 2020), and the Nyström method (Hataya & Yamada, 2023) along with matrix-vector products. Although this approach is efficient and used in large-scale problems (Choe et al., 2023; Hataya et al., 2022; Zhang et al., 2021), its application is limited to hyperparameters that directly change inner-level loss functions. In other words, the implicit-differentiation approach cannot be used for other hyperparameters, such as the learning rate of the inner-level optimizer $\Theta$.

The proposed glocal hypergradient estimation can rely on the unrolling approach but differentiating through only $\tau \; (\ll T)$ iterations like the local unrolling approach; thus, this approach is applicable to various hyperparameters as the unrolling approach.

## 2.3 KOOPMAN OPERATOR THEORY

Here, we roughly introduce the Koopman operator theory. For a complete introduction, refer to, for example, Brunton et al. (2022); Kutz et al. (2016).

Consider a discrete-time dynamical system on $\mathbb{R}^m$ represented by $f : \mathbb{R}^m \to \mathbb{R}^m$ such that $\boldsymbol{x}_{t+1} = f(\boldsymbol{x}_t)$ for $t \in \mathbb{Z}$. Then, given a measurement function $g : \mathbb{R}^m \to \mathbb{R}$ in some function space $\mathcal{G}$, the Koopman operator is defined as an infinite-dimensional linear operator $\mathcal{K}$ such that $\mathcal{K}g = g \circ f$. In other words, the Koopman operator advances via an observable $g$ the dynamical system one step forward: $\mathcal{K}g(\boldsymbol{x}_t) = g(f(\boldsymbol{x}_t)) = g(\boldsymbol{x}_{t+1})$.

Let $\mathcal{K}$'s eigenfunctions and eigenvalues be $\varphi_i : \mathbb{R}^m \to \mathbb{C}$ and $\lambda_i \in \mathbb{C}$. We suppose that $\mathcal{K}$ is invariant in $\mathcal{G} = \mathrm{span}\{\psi_1, \dots, \}$, i.e., $\mathcal{K}g \in \mathcal{G}$ for any $g \in \mathcal{G}$. Then, $g$ can be decomposed as $g = \sum_j v_j \psi_j$, where $v_j = \langle \psi_j, g \rangle \in \mathbb{C}$ is often referred to as a Koopman mode. Consequently, an observation at any time $t$ can be represented as

$$g(\boldsymbol{x}_t) = \mathcal{K}^{t-1} g(\boldsymbol{x}_0) = \mathcal{K}^{t-1} \sum_j \varphi_j(\boldsymbol{x}_0) v_j = \sum_j \lambda_j^{t-1} \varphi_j(\boldsymbol{x}_0) v_j. \tag{7}$$

For sufficiently large $t$, terms with $|\lambda_j| \neq 1$ diverge or disappear. As a result, if the dynamics involve no diverging modes, the steady state of $g(\boldsymbol{x}_t)$ can be written as

$$g(\boldsymbol{x}_\infty) \approx \sum_{j:|\lambda_j|=1} \lambda_j^{t-1} \varphi_j(\boldsymbol{x}_0) v_j. \tag{8}$$

Terms with $|\lambda_j| = 1$ but $\lambda_j \neq 1$ will oscillate in the state space, and those with $\lambda_j = 1$ will converge to a fixed point.

Numerically, we need to use a finite-dimensional matrix $\boldsymbol{K}$ to represent the operator $\mathcal{K}$. To do so, a set of measurement functions $\boldsymbol{g} := [g_1, \dots, g_n]^T$ ($g_i \in \mathcal{G}$) such that $g = \boldsymbol{c}^H \boldsymbol{g}$ ($\boldsymbol{c} \in \mathbb{C}^n$) is used so that $\mathcal{K}g \approx \boldsymbol{c}^H \boldsymbol{K} \boldsymbol{g}$ for any $g$. Such a matrix $\boldsymbol{K}$ is obtained by using (extended) dynamic mode decomposition (DMD, Kutz et al. 2016) that solves

$$\boldsymbol{K} := \underset{\boldsymbol{A} \in \mathbb{C}^{n \times n}}{\mathrm{argmin}} \sum_{t'=0}^{t-1} \| \boldsymbol{A} \boldsymbol{g}(\boldsymbol{x}_{t'}) - \boldsymbol{g}(\boldsymbol{x}_{t'+1}) \|_2^2. \tag{9}$$

The Koopman operator theory has demonstrated its effectiveness in the deep learning literature (Hashimoto et al., 2024; Konishi & Kawahara, 2023; Naiman & Azencot, 2023) and the analysis of optimization algorithms (Redman et al., 2022a; Dietrich et al., 2020). In particular, some works used it in the optimization of neural networks (Dogra & Redman, 2020; Manojlović et al., 2020; Šimánek et al., 2022) and network pruning (Redman et al., 2022b). However, these methods require DMD on the high-dimensional neural network parameter space, making its applications to large-scale problems difficult. Our method also relies on the Koopman operator theory, but we used DMD for the lower-dimensional hypergradients to indirectly advance the dynamics of neural network training, allowing more scalability.

## 3 GLOCAL HYPERGRADIENT ESTIMATION

As explained in Section 2.1, both global and local hypergradient have pros and cons. Specifically, global hypergradient can optimize the desired objective (Equation (3)), but it is computationally demanding. On the other hand, local gradient can be obtained efficiently, but it may diverge from the final objective.

This paper proposes *glocal* hypergradient that leverages the virtues of these contrastive approaches. Namely, glocal hypergradient approximates the global hypergradient from a trajectory local hypergradients using the Koopman operator theory to achieve

$$\boldsymbol{\phi}_{s+1} = \Phi(\hat{\boldsymbol{\theta}}_\infty^{(s)}, \boldsymbol{\phi}_s; \tilde{\mathcal{D}}) \quad \text{such that} \quad \boldsymbol{\theta}_t(\boldsymbol{\phi}) = \Theta(\boldsymbol{\theta}_{t-1}, \boldsymbol{\phi}_s; \mathcal{D}). \tag{10}$$

for $t \in \mathcal{I}_s$. $\hat{\boldsymbol{\theta}}_\infty^{(s)}$ is the estimate of the model parameter after the model training $\boldsymbol{\theta}_T$ so that $\nabla_\phi \tilde{\ell}(\hat{\boldsymbol{\theta}}_\infty^{(s)}(\boldsymbol{\phi})) \approx \nabla_\phi \tilde{\ell}(\boldsymbol{\theta}_T(\boldsymbol{\phi}))$ only by using local hypergradients obtained in $\mathcal{I}_s$. In the remaining

text, the superscript indicating the outer time step $(^{(s)})$ is sometimes omitted for brevity. Figure 1 schematically illustrates the proposed glocal hypergradient estimation with global and local hypergradients.

To approximate global hypergradient from a trajectory of local hypergradients, we use the Koopman operator theory. We regard the transition of local hypergradients during $\mathcal{I}_s$ as

$$\boldsymbol{h}_t := \nabla_{\boldsymbol{\phi}_s} \tilde{\ell}(\boldsymbol{\theta}_t) = \nabla_{\boldsymbol{\theta}} \tilde{\ell}(\boldsymbol{\theta}_t) \frac{\mathrm{d}\boldsymbol{\theta}_t}{\mathrm{d}\boldsymbol{\phi}_s}, \tag{11}$$

as a nonlinear dynamical system in terms of $t$. When using forward-mode automatic differentiation that computes Jacobian-matrix product $J_f : (\boldsymbol{x}, \boldsymbol{V}) \mapsto \partial f(\boldsymbol{x})\boldsymbol{V}$, where $f : \mathbb{R}^{n' \times m'}, \boldsymbol{x} \in \mathbb{R}^{n'}$, and $\boldsymbol{V} \in \mathbb{R}^{n' \times k'}$ for some $n', m', k' \in \mathbb{N}$, the right-hand side can be computed iteratively as

$$\underbrace{\frac{\mathrm{d}\boldsymbol{\theta}_t}{\mathrm{d}\boldsymbol{\phi}_s}}_{=: \boldsymbol{Z}_t} = \frac{\partial \Theta(\boldsymbol{\theta}_{t-1}, \boldsymbol{\phi}_s)}{\partial \boldsymbol{\theta}_{t-1}} \underbrace{\frac{\mathrm{d}\boldsymbol{\theta}_{t-1}}{\mathrm{d}\boldsymbol{\phi}_s}}_{= \boldsymbol{Z}_{t-1}} + \frac{\partial \Theta(\boldsymbol{\theta}_{t-1}, \boldsymbol{\phi}_s)}{\partial \boldsymbol{\phi}_s} = J_{\Theta(\cdot, \boldsymbol{\phi}_s)}(\boldsymbol{\theta}_{t-1}, \boldsymbol{Z}_{t-1}) + J_{\Theta(\theta_{t-1}, \cdot)}(\boldsymbol{\phi}_s, \boldsymbol{I}_q)$$

where $\boldsymbol{Z}_1 \in \mathbb{R}^{p \times q}$ is zero.

Following the literature of data-driven Koopman operator theory, we suppose that there exists a Koopman operator $\mathcal{K}$ and an observable $g$ that forwards this dynamical system one step as in Section 2.3, *i.e.*, $\mathcal{K}g(\nabla_{\boldsymbol{\phi}} \tilde{\ell}(\boldsymbol{\theta}_t)) = \mathcal{K}g(\boldsymbol{h}_t) = g(\boldsymbol{h}_{t+1}) = g(\nabla_{\boldsymbol{\phi}} \tilde{\ell}(\boldsymbol{\theta}_{t+1}))$ and $\mathcal{K}$ can be approximated by a finite-dimensional matrix $\boldsymbol{K} \in \mathbb{C}^{n \times n}$ using DMD with a set of measurement functions $\boldsymbol{g}$ (Mezić, 2005; Brunton et al., 2022). In the following discussion, we suppose $\boldsymbol{g} : \mathbb{R}^q \to \mathbb{R}^n$, $(q \leq n)$ be left invertible, that is, there exists $\boldsymbol{g}^\dagger$ such that $\boldsymbol{g}^\dagger g(\boldsymbol{x}) = \boldsymbol{x}$ for any $\boldsymbol{x} \in \mathbb{R}^q$.

Then, we can estimate the global hypergradient from local hypergradients:

$$\boldsymbol{g}(\boldsymbol{h}_T) \approx \boldsymbol{K}^{T-t} \sum_{j=1}^n b_j \boldsymbol{u}_j = \sum_{j=1}^n b_j \lambda_j^{T-t} \boldsymbol{u}_j, \tag{12}$$

where $t \in \mathcal{I}_s$, $\boldsymbol{u}_j \in \mathbb{C}^n$ is the $j$-th eigenvector with respect to the eigenvalue $\lambda_j \in \mathbb{C}$ of $\boldsymbol{K}$, and $b_j = \boldsymbol{u}_j^H \boldsymbol{g}(\boldsymbol{h}_t) \in \mathbb{C}$, *c.f.*, Equation (7).

If the spectral radius, the maximum norm of eigenvalues, is larger than 1, the global hypergradient will diverge, suggesting the current hyperparameters are in inappropriate ranges. Also, if there exists $k$ such that $|\lambda_k| = 1$ but $\lambda_k \neq 1$, the global hypergradient oscillates, suggesting the instability of the current hyperparameter choices. Thus, we need to assume that the spectral radius is not greater than 1 for the stability of hypergradient trajectories. Then, terms with $\lambda_j < 1$ can be ignored, as they will disappear for sufficiently large $\tau$. Indeed, this assumption holds in practical cases as shown in Section 4.3. Subsequently, we obtain *glocal* hypergradient

$$\hat{\boldsymbol{h}}_\infty := \nabla_{\boldsymbol{\phi}} \tilde{\ell}(\hat{\boldsymbol{\theta}}_\infty) = \boldsymbol{g}^\dagger \big( \sum_{j : \lambda_j = 1} b_j \boldsymbol{u}_j \big), \tag{13}$$

which approximates global hypergradients only from local information. We use this estimated hypergradient for updating hyperparameters using a gradient-based optimizer, such as vanilla gradient descent $\boldsymbol{\phi}_{s+1} = \Phi(\hat{\boldsymbol{\theta}}_\infty^{(s)}, \boldsymbol{\phi}_s; \tilde{\mathcal{D}}) = \boldsymbol{\phi}_s - \tilde{\eta} \hat{\boldsymbol{h}}_\infty^{(s)}$, where $\tilde{\eta}$ is a learning rate.

In summary, Algorithm 1 shows the pseudocode of the glocal hypergradient estimation with the setting that vanilla gradient descent is used for both inner and outer optimization.

## 3.1 COMPUTATIONAL COST

Remind that $p$ and $q$ are the numbers of model parameters and hyperparameters, respectively. Training neural networks for $\tau$ steps, as the function TRAINING in Algorithm 1, requires time complexity of $O(\tau p)$ and space complexity of $O(p)$ using the standard reverse-mode automatic differentiation. To compute the hypergradients, we have two options, *i.e.*, forward-mode automatic differentiation and reverse-mode automatic differentiation (Baydin et al., 2018). The first approach needs the evaluation of the Jacobian-matrix product in Line 5, resulting in time $O(q \cdot \tau p)$ and space $O(p + \tau q)$ complexities by repeating Jacobian-vector product. On the other hand, the reverse-mode approach involves $\tau$

---

**Algorithm 1** Pseudocode of the glocal hypergradient estimation

---

**input** Initialize $\boldsymbol{\theta}, \boldsymbol{\phi}$ and set $\boldsymbol{Z} = \boldsymbol{O}_{p \times q}$
1: %Model training for $\tau$ iterations
2: **def** TRAINING($\boldsymbol{\theta}, \boldsymbol{\phi}$):
3:     **for** $t$ in $1, 2, \ldots, \tau$ :
4:         $\boldsymbol{\theta} \leftarrow \boldsymbol{\theta} - \eta \nabla_{\boldsymbol{\theta}} \ell(\boldsymbol{\theta}, \boldsymbol{\phi}; \mathcal{D})$
5:         $\boldsymbol{Z} \leftarrow J_{\Theta(\cdot, \boldsymbol{\phi}_s)}(\boldsymbol{\theta}, \boldsymbol{Z}) + J_{\Theta(\boldsymbol{\theta}, \cdot)}(\boldsymbol{\phi}_s, \boldsymbol{I}_q)$
6:         $\boldsymbol{h}_t \leftarrow \nabla_{\boldsymbol{\theta}} \tilde{\ell}(\boldsymbol{\theta}, \tilde{\mathcal{D}}) \boldsymbol{Z}$
    **return** $[\boldsymbol{h}_1, \ldots, \boldsymbol{h}_\tau]$
7: %Hyperparamter update
8: **for** $s$ in $1, 2, \ldots, S$ :
9:     $[\boldsymbol{h}_1, \ldots, \boldsymbol{h}_\tau] \leftarrow$ TRAINING($\boldsymbol{\theta}, \boldsymbol{\phi}$)
10:     Compute $\hat{\boldsymbol{h}}_\infty$ from $[\boldsymbol{h}_1, \ldots, \boldsymbol{h}_\tau]$ using Equations (9) and (13)
11:     $\boldsymbol{\phi} \leftarrow \boldsymbol{\phi} - \tilde{\eta} \hat{\boldsymbol{h}}_\infty$

---

Table 1: Time and space complexities of gradient-based hyperparameter optimization with global, local, and glocal hypergradients using forward-mode automatic differentiation to compute hypergradients.

|  | Global | Local | Glocal |
|---|---|---|---|
| Time | $O(STpq)$ | $O(S\tau pq)$ | $O(S\tau pq + \min(n^2\tau, n\tau^2) + \min(n, \tau)^3)$ |
| Space | $O(p+q)$ | $O(p+q)$ | $O(p + \tau q + n^2)$ |

evaluations of TRAINING, computed in time $O(\tau \cdot (\tau p + q))$ and space $O(\tau(p+q))$ (Franceschi et al., 2017). In the experiments, we adopt forward-mode automatic differentiation to avoid reverse-mode's quadratic complexities with respect to $\tau$. However, when the number of hyperparameters $q$ is large, and $\tau$ is limited, reverse-mode automatic differentiation would be preferred. Additionally, the DMD algorithm at Line 10 requires space $O(n^2)$ and time $O(\min(n^2\tau, n\tau^2) + \min(n, \tau)^3)$ complexities, where $n$ is the number of observation functions and is $O(q)$ in most cases. Each term comes from singular value decomposition (SVD) to solve Equation (9) and eigendecomposition of $\boldsymbol{K}$.

Table 1 shows the complexities of gradient-based hyperparameter optimization with global, local, and glocal hypergradients, when forward-mode automatic differentiation is adopted for the outer-level problem. Because the computation of global hypergradients corresponds to the case where $\tau = T$, and it is used $S$ times, its time complexity is $O(STpq)$. Compared with the cost to evaluate the Jacobian, the additional cost by DMD is usually ignorable as it is independent of $p$. Thus, the proposed estimation is as efficient as computing a local hypergradient.

### 3.2 THEORETICAL PROPERTY

The error of the proposed glocal hypergradient from the actual global hypergradient is bounded as follows.

**Theorem 3.1.** *Assume that the dynamics of hypergradients in each $\mathcal{I}_s$, $[\boldsymbol{h}_t]_{t \in \mathcal{I}_s}$, is governed by a finite-dimensional Koopman operator $\bar{\boldsymbol{K}} \in \mathbb{C}^{n \times n}$, and this operator can be approximated with $\boldsymbol{K} \in \mathbb{C}^{n \times n}$ using DMD, and the spectral radii of $\bar{\boldsymbol{K}}$ and $\boldsymbol{K}$ are 1. Then,*

$$\|\boldsymbol{h}_T - \hat{\boldsymbol{h}}_\infty\|_2 \leq \|\boldsymbol{U}^{-1}\|_F \{\|\boldsymbol{e}_\tau\|_2 + (T - s\tau)\varepsilon_\tau\} + \sum_{j : |\lambda_j| < 1} |b_j| |\lambda_j|^\tau \|\boldsymbol{u}_j\|_2, \qquad (14)$$

*where $\boldsymbol{U} = [\boldsymbol{u}_1, \ldots, \boldsymbol{u}_q]$ consists of eigenvectors of $\boldsymbol{K}$, $\boldsymbol{e}_\tau = \boldsymbol{h}_{s\tau} - \boldsymbol{g}^\dagger(\boldsymbol{K}\boldsymbol{g}(\boldsymbol{h}_{s\tau-1}))$, and $\varepsilon_\tau$ is a constant only depends on the number of local hypergradients $\tau$.*

*Proof.* The left hand side of Equation (14) can be decomposed as $\|\boldsymbol{h}_T - \hat{\boldsymbol{h}}_\infty\|_2 \leq \|\boldsymbol{h}_T - \hat{\boldsymbol{h}}_T\|_2 + \|\hat{\boldsymbol{h}}_T - \hat{\boldsymbol{h}}_\infty\|_2$, where $\hat{\boldsymbol{h}}_T$ is obtained from the DMD algorithm as $\hat{\boldsymbol{h}}_T = \sum_j b_j \lambda_j^{T-s\tau} \boldsymbol{u}_j$. Then, we get

$$\|\boldsymbol{h}_T - \hat{\boldsymbol{h}}_T\|_2 \leq \|\boldsymbol{U}^{-1}\|_F \{\|\boldsymbol{e}_\tau\|_2 + (T - s\tau)\varepsilon_\tau\} \qquad (15)$$

by using Theorem 3.6 of Lu & Tartakovsky (2020) and

$$\|\hat{\boldsymbol{h}}_T - \hat{\boldsymbol{h}}_\infty\|_2 = \|(\sum_j - \sum_{j:|\lambda_j|=1}) b_j \lambda_j^{T-s\tau} \boldsymbol{u}_j\|_2 = \|\sum_{j:|\lambda_j|<1} b_j \lambda_j^{T-s\tau} \boldsymbol{u}_j\|_2 \le \sum_{j:|\lambda_j|<1} |b_j| |\lambda_j|^\tau \|\boldsymbol{u}_j\|_2, \tag{16}$$

since $T - s\tau \ge \tau$.

*Remark* 3.2. The first term of Equation (14) decreases as the outer optimization step $s$ proceeds. In this term, $\|\boldsymbol{e}_\tau\|$ is minimized by the DMD algorithm (Equation (9)). $\varepsilon_\tau$ and the second term of Equation (14) decrease as we use more local hypergradradients $\tau$ for estimation, because $\boldsymbol{K}$ converges to $\bar{\boldsymbol{K}}$ as $\tau$ increases (Korda & Mezić, 2018). When $T$ is constant, increasing $\tau$ improves the quality of the glocal hypergradient estimation while decreasing the number of the outer step, and we need to take their trade-off in practice.

## 4 EXPERIMENTS

This section empirically demonstrates the effectiveness of the glocal hypergradient estimation. The source code to reproduce the experiments will be publicly released upon publication.

**Implementation** We implemented neural network models and algorithms, including DMD, using JAX (v0.4.28) (Bradbury et al., 2018), optax (DeepMind et al., 2020), and equinox (Kidger & Garcia, 2021). Reverse-mode automatic differentiation is used for model gradient computation. Forward-mode automatic differentiation is adopted for hypergradient computation.
In the following experiments, we adopt Hankel DMD (Arbabi & Mezic, 2017), whose observation function $\boldsymbol{g}$ can be defined using time-delayed coordinates as $\boldsymbol{g}(\boldsymbol{h}_t) = [\boldsymbol{h}_t; \boldsymbol{h}_{t+1}; \ldots; \boldsymbol{h}_{t+m-1}] \in \mathbb{R}^{mq}$ and $t = s\tau, \ldots, s\tau - m + 1$ for some positive integer $m$ for each $\mathcal{I}_s$. Such a function $\boldsymbol{g}$ is obviously left-invertible.

**Setup** For DMD, we use the last $\sigma$ hypergradients out of $\tau$ hypergradients in each $\mathcal{I}_s$ because the dynamics immediately after updating hyperparameters may be unstable. Throughout the experiments, we optimize model parameters 10k times and hyperparameters every 100 model updates, *i.e.*, $\tau = 100$, using Adam optimizer with a learning rate of 0.1 (Kingma & Ba, 2015). We set $m = 10$ and $\sigma = 80$, except for the analysis in Section 4.3. When computing hypergradients, we use hold-out validation data $\tilde{\mathcal{D}}$ from the original training data, following Micaelli & Storkey (2021). To minimize the effect of stochasticity, we use as large a minibatch size as possible for the validation. Average performance on test data over three different random seeds is reported. Further experimental details can be found in Appendix B.

**Baselines** We adopt local and global baselines that greedily update hyperparameters using global and local hypergradients, respectively. As the global baseline is prone to diverge and computationally expensive, we can only present its results on MNIST-sized datasets.

### 4.1 OPTIMIZING OPTIMIZER HYPERPARAMETERS

Appropriately selecting and scheduling hyperparameters of optimizers, in particular, learning rates, is essential to the success of training machine learning models (Bergstra et al., 2011). Here, we demonstrate the validity of the glocal hypergradient estimation in optimizing such hyperparameters. Unlike (Micaelli & Storkey, 2021), we adopt forward-mode automatic differentiation to differentiate through the optimization steps, which allows to use any differentiable optimizers, including SGD and Adam, for the inner optimization without requiring any manual reimplementation.

**LeNet on MNIST variants** First, we train LeNet (LeCun et al., 2012) using SGD with learnable learning rate, momentum, and weight-decay rate hyperparameters, and Adam (Kingma & Ba, 2015) with learnable learning rate and betas. The logistic sigmoid function is applied to these hyperparameters to limit their ranges in $(0, 1)$. MNIST variants, namely, MNIST (Le Cun et al., 1998), Kuzushiji-MNIST (KMNIST, Clanuwat et al. (2018)) and Fashion-MNIST (FMNIST, Xiao et al. (2017)) are used as datasets. The LeNet has 15k parameters and is trained for 10k iterations.
Figures 2 and C.3 shows learning curves with the transition of hyperparameters of optimizers on the FMNIST dataset. We can observe that the development of hyperparameters of the glocal approach exhibits similar trends with the global baseline, and the glocal and global approaches show nearly identical performance. Contrarily, the local baseline changes hyperparameters aggressively, resulting

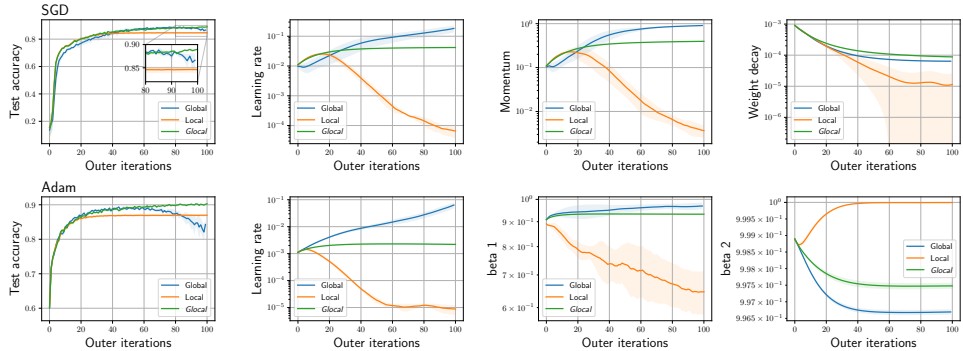

Figure 2: Test accuracy and the transition of the hyperparameters of SGD and Adam. The proposed local approach shows similar hyperparameter development to the global baseline.

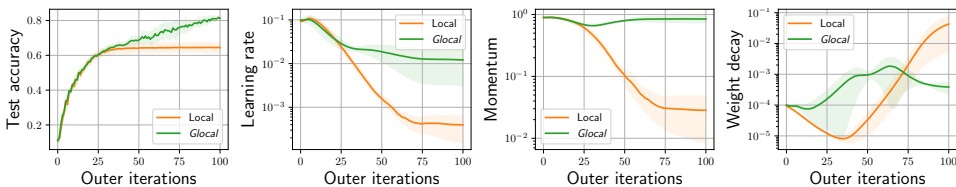

Figure 3: Test accuracy curves and the transition of SGD's hyperparameters of WideResNet 28-2 on CIFAR-10.

in suboptimal performance. The glocal approach successfully avoids the short-horizon bias of the local approach. Additionally, the global baseline shows performance degeneration at the end of the training, which may be attributed to gradient explosion/vanishing. The proposed method can circumvent this issue. Other results can be found in Appendix C.

**WideResNet on CIFAR-10/100 and SVHN** Next, we train WideResNet 28-2 (Zagoruyko & Komodakis, 2016) on CIFAR-10/100 Krizhevsky et al. (2009) and SVHN Netzer et al. (2011) using SGD with learnable learning rate and weight-decay rate hyperparameters. The used model has 1.5M parameters and is trained for 10k iterations. The results are presented in Figures 3, C.4 and C.5 Contrarily to the local baseline drastically changing hyperparameters, the glocal estimation adjusts the hyperparameters gradually and yields better performance, partially because it can predict the future state.

## 4.2 DATA REWEIGHTING

Data reweighting task trains a meta module $\mu_\phi$ to reweight a loss value to each example to alleviate the effect of class imbalanceLi et al. (2021); Shu et al. (2019). $\mu_\phi : \mathbb{R} \to (0, 1)$ is an MLP, and the inner loss function of the task is $\ell(\boldsymbol{\theta}; \mathcal{D}) = \sum_{(\boldsymbol{x},y) \in \mathcal{D}} L(\boldsymbol{x}, y; \boldsymbol{\theta}) \cdot \mu_\phi(L(\boldsymbol{x}, y; \boldsymbol{\theta}))$, where $L : \mathcal{D} \to \mathbb{R}$ is cross-entropy loss. $\phi$ is trained on balanced validation data.

We train WideResNet 28-2 on imbalanced CIFAR-10 and CIFAR-100 Cui et al. (2019), which simulate class imbalance. Specifically, the imbalanced data with an imbalance factor of $f$ reduces the number of data in the $c$-th class to $\lfloor \frac{1}{f^{c/C}} \rfloor$, where $C$ is the number of categories, *e.g.*, 10 for CIFAR-10. As the meta module, we adopt a two-layer MLP with a hidden size of 128, consisting of 385 parameters. Table 2 demonstrates that the glocal approach surpasses the local baseline, indicating its effectiveness to a wide range of problems.

## 4.3 ANALYSIS

Below, we analyze the factors of the glocal hypergradient estimation using the task in Section 4.1 on the FMNIST dataset.

**Eigenvalues of DMD** In Section 3, the existence of eigenpairs whose eigenvalues equal to one was assumed for efficient approximation of a global hypergradient. To see whether it holds in practice, the

Table 2: Test accuracy of WideResNet 28-2 trained on imbalanced datasets.

| Dataset / Imbalance Factor | Local | *Glocal* |
|---|---|---|
| CIFAR-10 / 50 | 0.520 | 0.715 |
| CIFAR-10 / 100 | 0.445 | 0.595 |
| CIFAR-100 / 50 | 0.315 | 0.350 |

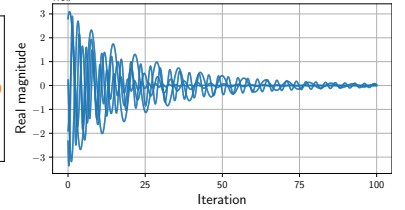 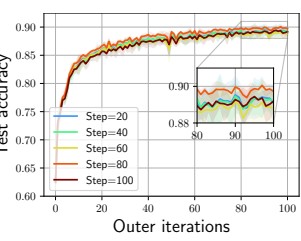

Figure 4: **Left**: The eigenvalues obtained by the Hankel DMD for a hypergradient trajectory. Eigenvalues nearly close to 1 are highlighted in orange. **Middle**: The magnitude of modes of the estimated hypergradient corresponding to the learning rate $b_j \lambda_j^t u_j$ for $j : \lambda_j \neq 1$ over 100 iterations of $t$. **Right**: The comparison of validation performance of the proposed estimation with different configurations. LeNet is trained on FMNIST with an initial learning rate of $0.1$.

Table 3: Runtime comparison of optimizing optimizer hyperparameters with LeNet in second. "w/o HPO" indicates training without hyperparameter optimization process.

| w/o HPO | Local | Global | *Glocal* |
|---|---|---|---|
| 11.4 | 58.2 | 2891.2 | 58.3 |

left panel of Figure 4 shows the eigenvalues obtained by the Hankel DMD at $s = 1$. We can observe two eigenvalues nearly close to 1 (highlighted in orange). Additionally, the middle panel of Figure 4 illustrates that modes with other eigenvalues decay rapidly. Because the magnitudes of the modes associated with the eigenvalue of 1 is an order of $10^{-3}$, other modes can be ignored, numerically supporting the validity of our assumption.

**DMD Configurations** We compare the proposed method with different configurations to see how the configurations of the Hankel DMD algorithm, specifically, the number of hypergradients ($\sigma$, the right panel of Figure 4) and the number of stacks per column ($m$, Figure C.7), affect the estimation. We can observe that it performs better when DMD uses the last 80 hypergradients out of 100. Although the estimation is expected to improve as more hypergradients are used, they may be unstable after the change of hyperparameters, which degenerates the quality of the prediction; discarding the first 20 hypergradients may avoid such an issue.

**Runtime** Table 3 compares runtime on a machine equipped with an AMD EPYC 7543P 32-Core CPU and an NVIDIA RTX 6000 Ada GPU with CUDA 12.3. Note that because of the limitation of JAX that eigen decomposition for asymmetric matrices on GPU is not yet supported, the computation of the adopted DMD algorithm is suboptimal. Nevertheless, the proposed method achieves significant speedup compared with the global baseline, revealing its empirical efficiency.

## 5 DISCUSSION AND CONCLUSION

This paper introduced *glocal* hypergradient estimation, which leverages the virtues of global and local hypergradients simultaneously. To this end, we adopted the Koopman operator theory to approximate a global hypergradient from a trajectory of local hypergradients. The numerical experiments demonstrated the validity of gradient-based hyperparameter optimization using glocal hypergradients.

**Limitations** In this work, we have implicitly assumed that the meta criterion converges after enough iterations, and so do hypergradients. This assumption may be strong when a certain minibatch drastically changes the value of the meta criterion, such as adversarial learning. Luckily, we did

not suffer from such a problem, probably because we focused our experiments only on supervised learning with sufficiently sized datasets and used full-batch validation data. Introducing stochasticity by the Perron-Frobenius operator (Korda & Mezić, 2018; Hashimoto et al., 2020), an adjoint of the Koopman operator, would alleviate this limitation, which we leave for future research.

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

## A    ADDITIONAL DISCUSSION ON THEOREM 3.1

The first half of the proof of Theorem 3.1 depends on the theorem 3.6 of Lu & Tartakovsky (2020), which shows Equation (15) without equality holds if $\bar{K}$'s spectral radius $\rho(\bar{K}) < 1$. This condition can be relaxed to $\rho(\bar{K}) \leq 1$, and then Equation (15) holds. $\varepsilon_\tau$ in Theorem 3.1 can be given as

$$\varepsilon_\tau = \left\{ c_\tau \left( \|\boldsymbol{h}_1\|_2^2 + \sum_{t=1}^\tau \|\boldsymbol{f}_t\|_2^2 \right) \right\}^{1/2},\tag{17}$$

where $\boldsymbol{f}_t = \bar{K}\boldsymbol{g}(\boldsymbol{h}_t) - \boldsymbol{g}(\boldsymbol{h}_{t+1})$ and $c_\tau \geq \|\bar{K} - K\|_2^2$ depends on the number of hypergradients.

## B    DETAILED EXPERIMENTAL SETTINGS

Throughout the experiments, the batch size of training data was set to 128.

### B.1    OPTIMIZING OPTIMIZER HYPERPARAMETERS

**LeNet** We set the number of filters in each convolutional layer to 16 and the dimension of the following linear layers to 32. The leaky ReLU is used as its activation.

**WideResNet** We modified the original WideResNet 28-2 in Zagoruyko & Komodakis (2016) as follows: replacing the batch normalization with group normalization Wu & He (2018) and adopting the leaky ReLU as the activation function.

**Validation data** We separated 10% of the original training data as validation data.

### B.2    DATA REWEIGHTING

**WideResNet** We modified the original WideResNet 28-2 in Zagoruyko & Komodakis (2016) as follows: replacing the batch normalization with group normalization and adopting the leaky ReLU as the activation function. The model was trained with an inner optimizer of SGD with a learning rate of 0.01, momentum of 0.9, and weight decay rate of 0.

**Valiation data** We separated 1000 data points from the original training dataset to construct a validation set.

## C    ADDITIONAL EXPERIMENTAL RESULTS

Figures C.1 and C.2 present the test accuracy curves and the transition of SGD's hyperparameters of KMNIST and MNIST. As the results presented in the main text in Section 4.1, the proposed glocal approach yields similar behaviors as the global one, while maintaining the efficiency of the local approach. As can be seen from Figures C.2 and C.3, the global method often fails because of loss explosion, as discussed in the literature (Micaelli & Storkey, 2021). Similarly, Figure C.4 shows the results on the CIFAR-100 dataset and SVHN.

Figure C.7 shows test accuracy and the transition of SGD hyperparameters (learning rate, momentum, weight decay) with different numbers of stacks of the Hankel DMD ($m$). When $m = 1$, equivalent to the case when $\boldsymbol{g}$ is identity, the estimated hyperparameters yield much higher variances. This can be explained as the error of approximating the nonlinear dynamical system of hypergradients with a linear dynamical system in the space of hypergradients. Contrarily, the hyperparameters obtained with Hankel DMD with $m > 1$ show limited variances, indicating that Hankel DMD could more appropriately capture the nonlinear dynamics with lower variances.

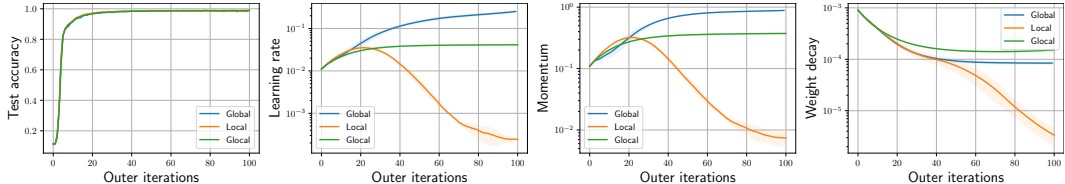

Figure C.1: The transition of the SGD's hyperparameters and test accuracy curves of LeNet on MNIST with an initial learning rate of 0.01.

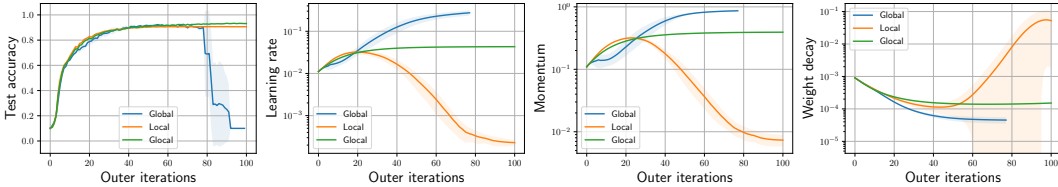

Figure C.2: The transition of the SGD's hyperparameters and test accuracy curves of LeNet on KMNIST with an initial learning rate of 0.01.

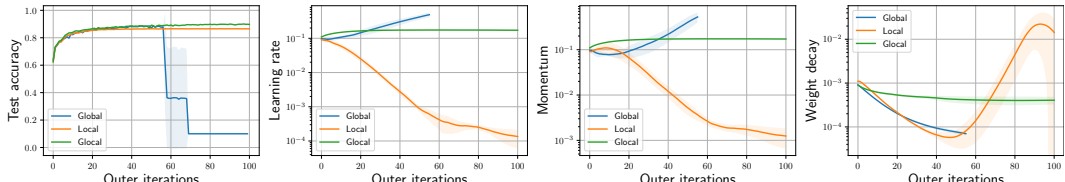

Figure C.3: The transition of the SGD's hyperparameters and test accuracy curves of LeNet on FMNIST with an initial learning rate of 0.1.

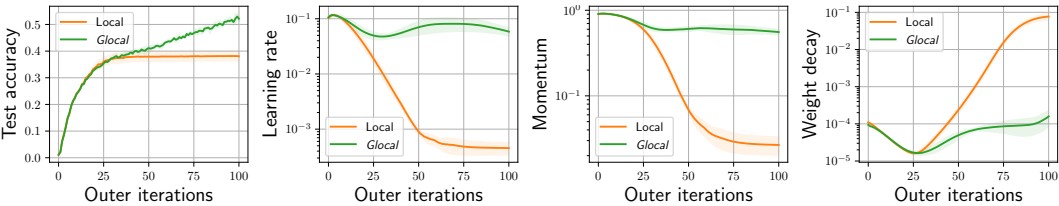

Figure C.4: The transition of the SGD's hyperparameters and test accuracy curves of WideResNet 28-2 on CIFAR-100.

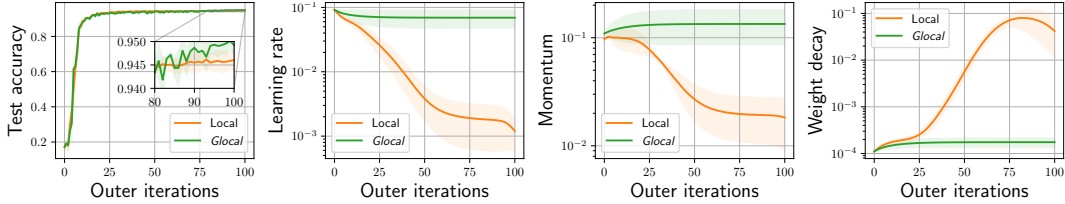

Figure C.5: The transition of the SGD's hyperparameters and test accuracy curves of WideResNet 28-2 on SVHN.

Figure C.7 shows the test accuracy curves and the transition of SGD's hyperparameters of FMNIST with the global baseline method, its truncated approximation, and the proposed method. Specifically, the approximation method truncates the computation of the global baseline after 100 iterations. We can observe that the proposed glocal method shows trends that are more similar to the global one than the truncated approximation.

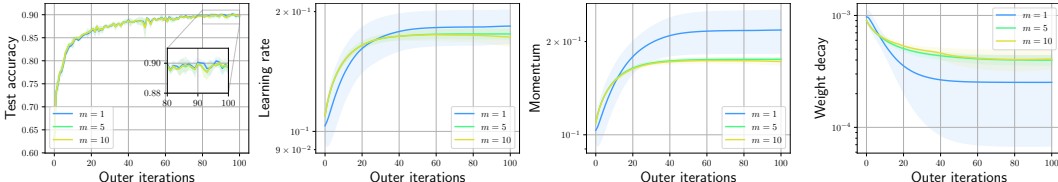

Figure C.6: Test accuracy and the transition of hyperparameters of SGD with different numbers of stacks of the Hankel DMD ($m$). $m = 1$ corresponds to the vanilla DMD.

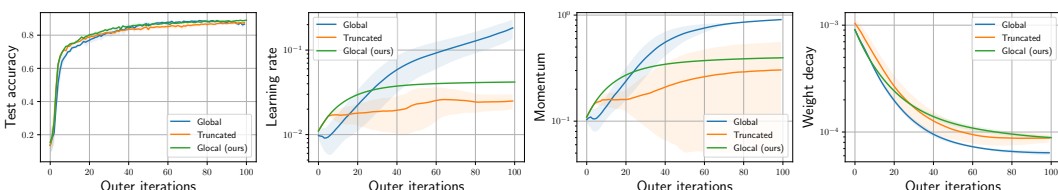

Figure C.7: Test accuracy and the transition of hyperparameters of SGD with the global method, its truncated approximation, and the proposed method.

