# OpenReview forum: "Glocal Hypergradient Estimation with Koopman Operator"
_ICLR.cc/2025/Conference — Submitted to ICLR 2025_

### Official Review · Reviewer_h2T9 · 2024-10-31

**Soundness:** 3
**Presentation:** 3
**Contribution:** 2
**Rating:** 5
**Confidence:** 3

**Summary:**

This work considers the problem of hyperparameter optimization for bilevel optimization problems. By leveraging Koopman operator theory, the authors provide a method to estimate global hypergradients (usually only available after fully solving the inner problem) with local hypergradient trajectories (partial solution trajectories of the inner prolbem).

**Strengths:**

I'm on the fence on this submission -- the method is well-motivated and the derivation is for the most part clear, but I felt that the experimental results are somewhat of a let down.

* The paper is overall well-written, with a few minor typos. The authors do an admirable job of making their theory tractable and easy to read.
* Meta optimization is an important problem and the research setting is well-motivated.
* The authors provide a thorough runtime comparison and associated discussion, which shows that their approach is as computationally efficient as using local hypergradient.

**Weaknesses:**

1. I'm somewhat suspicious of the handwaving around non-unit eigenvalues. Specifically, consider the section from line 246 - 252, where basically all eigenvalues besides those which are equal to one are discarded. Is there any theoretically grounded explanation from doing so? If the koopman operator says that the global hypergradients should oscillate, when intervene and artificially eliminate those modes? Similarly, when solving the DMD for $K$ as in (9), I don't see why you would get modes with an eigenvalue exactly equal to one as there is random noise in the collected hypergradient data.
2. The experiment settings are minimal, and only involve relatively simple image classification tasks.
3. The experimental results somewhat conflict with the aims of the paper. Namely, the performance of the global hyperparameter estimation degrades severely in the bottom-left panel of Figure 2. In Figures C.2 and C.3 in the appendix are even worse. The appendix briefly mentions "loss explosion," but a further discussion is warranted, since my impression is that the whole point of this paper is to provide a fast way of approximating the "gold-standard" global hypergradients!
4. Additional baselines of estimating the global hypergradient should be considered. At least: simply using the hypergradient at iteration $\tau$ as a substitute for $h_T$ (just once, not continuing to train as in local hypergradient estimation).

**Questions:**

Comments & questions
1. I suggest mentioning, at least in passing, the initialization of $\theta_0$ beneath equation (4).
2. Minor notational note: in your problem formulation, you are adopting the convention of outer-level symbol carrying an additional tilde. The parameters are the exception ($\theta$ vs $\phi$); however, for the optimization algorithm, you are using capital $\Theta$ and $\tilde \Theta$. I like having separate symbols for the parameters, so for consistency I would recommend using a capital $\Phi$ instead of $\tilde \Theta$ for the outer-level gradient step. Or find a non-theta symbol to use to denote an optimization step. Just a suggestion, I'll leave it up to the authors.
3. In equation (8), why do terms which diverge ($|\lambda_j| > 1$) no longer appear in the expression? I would think that they would dominate all other modes.
4. For what trajectory $x_t$ is (9) being solved? Wouldn't it make sense to look at many trajectories from different initial conditions?
5. I'm confused about the implications of the bound in Theorem 3.1. The bound aggregates terms corresponding to non-unit eigenvalues, presumably becasue these were discarded previously (see Weakness 1). Wouldn't your theorem suggest that including these terms would result in less estimation error?
6. I don't understand the measurement defined in line 345. How can the measurement function depend on future hypergradients, some of which potentially haven't been computed yet? For example, how would you compute $g(h_{t^*})$ where $t^*$ is the biggest index in $\mathcal{I}_s$?

Minor notes:
* Line 123: "global hypergardient"
* Line 167: the inputs and outputs of $g$ should be clarified; I presume $g: R^m \to R$
* Line 172: mixing up $\phi$ and $\varphi$
* Line 234: "as in Section 2.3"
* Line 334: "outer steps"
* Figure 4: periods after left/middle/right

---

> ### Author Response · Authors · 2024-11-24
>
> Thank you for the insightful comments. Below, we answer your questions and concerns.
>
> ---
>
> - W1: The existence of unit eigenvalues is derived from the assumption of the stable hypergradient after a long period of training. In the experiments, we tried to make the dynamics less noisy by using larger batch sizes.
> - W2: We adopted simple image classification tasks to make the dynamics less stochastic.
> - W3: The global method inevitably accumulates numerical errors over long-horizon, causing gradient explosion or vanishing (L149), similar to the training of pure RNNs. Since the proposed method relies on short-horizon information, it successfully avoids this issue. We will discuss this problem more carefully.
> - W4: Thank you for suggesting another baseline that truncates the global method. We included the comparison in Fig C6 of the updated manuscript.
>
> ---
>
> - Q1, Q2: Thank you for the comments. We updated the manuscript accordingly.
> - Q3: We fixed the explanation as “if the dynamics involve no diverging modes” and the equation as $g(x_\infty)\approx \sum_{j: |\lambda_j|=1}\lambda_j^{t-1} \varphi_j(x_0) v_j.$
> - Q4: $x_0, \dots, x_t$ is a trajectory of a dynamical system $x_{t+1}=f(x_t)$.
> - Q5: In Eq 8, we discuss the state after convergence. For finite time steps $T$, the effect of modes with eigenvalues smaller than 1 is not negligible.
> - Q6: This measurement function is applied to a batch of observations $[h_0, \dots, h_\tau]$ and converts it to $[k_0, \dots, k_{\tau-m+1}]$, where $k_t=[h_t,\dots,h_{t+m-1}]$. Thus, we do not need to compute $g(h_{t^*})$, where $t^*$ is the biggest index in $\mathcal{I}_s$. We updated the description accordingly.

---

### Official Review · Reviewer_bcjq · 2024-11-03

**Soundness:** 2
**Presentation:** 2
**Contribution:** 3
**Rating:** 5
**Confidence:** 3

**Summary:**

This paper proposes a novel method called glocal hypergradient estimation for hyperparameter optimization. This method combines the computational efficiency of local hypergradients with the reliability of global hypergradients. It utilizes Koopman operator theory to approximate global hypergradients from the trajectory of local hypergradients, enabling efficient and effective hyperparameter updates. Finally, the paper validates the effectiveness of the proposed method in hyperparameter optimization through experiments.

**Strengths:**

1. The integration of Koopman operator theory to enhance hypergradient estimation is a novel approach, offering a fresh perspective on hyperparameter optimization.

2. The method significantly reduces computational costs compared to traditional global hypergradient approaches.

3. The approach is scalable to large-scale problems, making it applicable to real-world deep learning tasks. Furthermore, the paper provides numerical experiments demonstrating the method's effectiveness in various scenarios.

**Weaknesses:**

1. Algorithm 1 and Theorem 3.1 rely on assumptions about the spectral radius and stability, which may not hold in all cases.

2. The theoretical foundation involving Koopman operators may be complex for practitioners unfamiliar with the concept.

3. The experiments are somewhat limited. Could additional datasets be included, or could comparative experiments be conducted on other models as well?

4. The presentation of the experimental results is somewhat unclear. For example, all the experimental results in Section 4.2 are only summarized in Table 2, with no accompanying figures. Could Table 3 offer a more detailed explanation?

**Questions:**

1. In equation (9), could you provide an analysis of the computational complexity of solving the optimization problem w.r.t $A\in \mathbb{C}^{n\times n}$? Furthermore, are there any potential challenges in its implementation?

2. In equation (13), the computation of $g^\dagger$ involves solving a system of linear equations $g^\dagger g(x) = x$. How does the computation affect the overall complexity of Algorithm 1?

3. In Section 4.1, why is the Global curve missing in Figure 3? Could you discuss any implications this might have for the comparison between methods?

4. How does the proposed approach compare with specific state-of-the-art hyperparameter optimization methods, such as Bayesian optimization or evolutionary algorithms, in terms of performance metrics like accuracy or convergence speed, as well as computational overhead?

There are some typos:
1. In equation (1), the meta-level function is expressed as $\tilde{l}(\theta^*(\Phi);\tilde{D})$. However, the meta objective in line 37 is written as  $\tilde{l}(\theta,\Phi;\tilde{D})$

2. In equation (9), the expression $\sum\limits_{t=0}^{t-1}$ has a conflict because the variable $t$ is used both as the index and the limit of summation.

3. In line 266, "requires time $O(\tau p)$ and space $O(p)$ complexities" should be "requires time complexity of $O(\tau p)$ and space complexity of $O(p)$".

4. The sentence in line 325 seems to be awkwardly structured.

---

> ### Author Response · Authors · 2024-11-24
>
> Thank you for the helpful review. Below is our rebuttal to your questions and concerns.
>
> ---
>
> W1: As the reviewer pointed out, this assumption does not always hold. However, it is generally used in many practical data-driven dynamical systems in various domains, and we followed it.
> - W2: We believe the concept is simple: the core idea is to use the least square (Eq 9) to estimate a matrix that forwards the linearized dynamics of hypergradient. Practitioners can use tools to implement DMD, such as pyDMD.
> - W3: Thank you for suggesting other experiments. We will add a new dataset. For models, we believe that LeNet and ResNet are sufficient to demonstrate the effectiveness of the proposed method on the datasets we used.
> - W4: It would be appreciated if you elaborated more on what makes tables 2 and 3 unclear.
>
> ---
>
> - Q1: We analyzed the complexities of DMD in Section 3.1 (around L298).
> - Q2: If we augment the observation function as $(x, \mathbf{g}(x))$, then its left inverse is simply taking the first $\dim x$ elements, whose computational cost is negligibly small. Hankel DMD holds this property.
> - Q3: The global method takes much longer computation than others, as explained in Table 3. Thus, we omit the results on non-MNIST datasets, as explained in L358.
> - Q4: It is difficult to compare the gradient-based HPO methods with black-box ones directly. In terms of complexities, gradient-based ones are much more efficient because they can additionally use gradient information.

---

> > ### Author Response · Authors · 2024-11-26
> > **Update the answer to W3**
> >
> > We additionally conducted experiments of optimization of the optimizer's hyperparameters on the SVHN dataset (Section 4.1).
> > The proposed method shows slightly better results than the local baseline.

---

> > > ### Comment · Reviewer_bcjq · 2024-11-26
> > >
> > > Thanks for your patient response. After carefully considering your answer, we decide to raise the rating to 5 for your work.

---

> > > > ### Author Response · Authors · 2024-12-03
> > > >
> > > > Thank you for your consideration.

---

### Official Review · Reviewer_qy4S · 2024-11-04

**Soundness:** 2
**Presentation:** 2
**Contribution:** 2
**Rating:** 5
**Confidence:** 3

**Summary:**

This paper proposes a hypergradient descent method that uses local hypergradients to estimate the global hypergradient. The method is based on Koopman operator theory which uses a finite-dimensional linearization of the nonlinear dynamics of the hypergradient. Authors characterize the estimation error and provides detailed comparisons on the computational complexities of Glocal HGD, local HGD, and global HGD. Finally, they present experiments that show the practical efficacy of the method.

**Strengths:**

The paper studies an important problem, since hyperparameter optimization is a common challenge in training neural nets. The paper's approach using Koopman operator theory is clever and connects hyperparameter optimization to nonlinear dynamical systems. The algorithm and the computational complexities are clearly written, and I appreciate the diagnostic plots in the experimental section.

**Weaknesses:**

1. Important design choices are not given: (1) how should we select the dimension of the Koopman operator $n$? Intuitively, $n$ should depend on properties of the underlying dynamical system, and it would be helpful to have some guidelines. (2) how should we select $\textbf{g}$? Authors use Hankel DMD in the experiments, and it would be great to provide some justification.

2. Experiments: (1) the experiments are relatively small scale. Authors mention that global hypergradients are difficult to obtain for larger models, but still it would be good to compare Glocal to other baselines, for example the best tuned SGD. Without larger scale experiments, it is difficult to know how the proposed method scales with model size. (2) it would be useful to compare Glocal with the best tuned SGD/Adam as a baseline (3) it would be great to verify empirically that Glocal can indeed estimate the global hypergradient. If I understand correctly, one can compare the estimated gradient with the true global hypergradient.

3. Theoretical analysis: Theorem 3.1 assumes a Koopman operator of dimension $n$ exists. Without specifications on $n$, this result is less meaningful as $n$ can be very large. However, a large $n$ can make the Glocal method impractical. It would be helpful to provide justification and guidance on $n$.

4. Section 2.3 is not clear: (1) Line 172, what is the relationship between $\phi$ and $\varphi$? The decomposition of $g$ is written in $\phi$ on line 172 but in $\varphi$ in eq. (7). (2) Eq. (8) is not precise, because large $\lambda$'s will diverge, but the arrow notation usually means convergence. (3) The notation for the set of measurement functions $\textbf{g}$ and the individual functions $g_i$ is confusing because they can be very different from the original function $g$ and yet use the same letter.

**Questions:**

1. Can the proposed method handle changing dynamics, given that the training dynamics change over time (initially the loss drops significantly and then less so)?
2. For experiments, how do the results change over different initial learning rates? Can Glocal HGD always converge to a good learning rate no matter what the initial learning rate is?
3. Why does Glocal outperform global HGD towards the end of training?

Also see Weaknesses above.

---

> ### Author Response · Authors · 2024-11-24
>
> Thank you for the insightful comments. Below, we answer your questions and concerns.
>
> ---
>
> - W1: How to set the dimension of an approximated Koopman operator $n$ and the dictionary $\mathbf{g}$ is problem dependent. In our experiments, we adopt Hankel DMD stacking 10 time-delayed vectors so that $n=10q$, where $q$ is the number of hyperparameters.
> - W2: Thank you for suggesting experiments. Since the proposed method is an online approach, it is hard to outperform or even match the performance of the best-tuned ones obtained through countless trials and errors by researchers for decades. In the updated manuscript, we added another baseline (Fig C6) to demonstrate that the proposed method shows better approximation than a truncated variant of the global baseline.
> - W3: $n$ can be specified as written above.
> - W4: Thank you for carefully reading our manuscript. $\phi$ is a typo of $\psi$, and we fixed it. We updated the description of Eq8. The connection between $\mathbf{g}$ and $g$ is explained around L186.
>
> ---
>
> - Q1: The training dynamics can be written as $\theta_{t+1}=\Omega(\theta_t)$, where $\theta_t$ is model parameters and $\Omega$ is the optimizer, so in theory it is possible to handle the dynamics. In practice, on the other hand, it would be challenging to accurately predict the dynamics after a significant loss drop only by using trajectory before the event without enough prior knowledge of the dynamics. However, the predicted states from dynamics before loss drop could help us to determine if the initial hyperparameters would lead to divergence of training dynamics or not.
> - Q2: Figure C3 shows the results when the initial learning rate is 0.1.
> - Q3: Global HGD suffers from numerical instability, which appears after around 100 iterations.

---

### Meta-Review · Area_Chair_cBRF · 2024-12-18

**Metareview:**

This paper proposes a novel method called glocal hypergradient estimation for hyperparameter optimization. It uses Koopman operator theory to approximate global hypergradients from the trajectory of local hypergradients, enabling efficient and effective hyperparameter updates.

The reviewers generally agreed that the paper was well-written and presented a novel approach to an important problem.  They also found the theoretical contributions to be sound and the experimental results to be promising.  However, there were some concerns about the clarity of the presentation and the limited scope of the experiments.

Overall, there was consensus among the reviewers that the empirical results are not quite compelling enough to justify publication yet.

**Additional Comments On Reviewer Discussion:**

Reviewers raised concerns about parameter choices, experimental scale, theoretical assumptions, and result clarity. The authors clarified their methods, added experiments, and improved the manuscript.

---

### Decision · Program_Chairs · 2025-01-22

Reject